# Overexpression of the Aldehyde Dehydrogenase Gene *ZmALDH* Confers Aluminum Tolerance in *Arabidopsis thaliana*

**DOI:** 10.3390/ijms23010477

**Published:** 2022-01-01

**Authors:** Han-Mei Du, Chan Liu, Xin-Wu Jin, Cheng-Feng Du, Yan Yu, Shuai Luo, Wen-Zhu He, Su-Zhi Zhang

**Affiliations:** 1Key Laboratory of Biology and Genetic Improvement of Maize in Southwest China of Agricultural Department, Ministry of Agriculture, Maize Research Institute, Sichuan Agricultural University, Chengdu 611130, China; duhanmei1027@163.com (H.-M.D.); liuchan0910@163.com (C.L.); jinxinwu121419@163.com (X.-W.J.); duyeye1936551448@163.com (C.-F.D.); yuyanbigfish@163.com (Y.Y.); lorenzo212@163.com (S.L.); 2Panxi Crops Research and Utilization Key Laboratory of Sichuan Province, Xichang University, Xichang 615000, China; 3Crop Research Institute, Sichuan Academy of Agricultural Sciences, Chengdu 610066, China; wenzu_he@163.com

**Keywords:** Al toxicity, oxidative stress, *ZmALDH*, ascorbate-glutathione cycle, antioxidant enzymes

## Abstract

Aluminum (Al) toxicity is the main factor limiting plant growth and the yield of cereal crops in acidic soils. Al-induced oxidative stress could lead to the excessive accumulation of reactive oxygen species (ROS) and aldehydes in plants. Aldehyde dehydrogenase (*ALDH*) genes, which play an important role in detoxification of aldehydes when exposed to abiotic stress, have been identified in most species. However, little is known about the function of this gene family in the response to Al stress. Here, we identified an *ALDH* gene in maize, *ZmALDH*, involved in protection against Al-induced oxidative stress. Al stress up-regulated *ZmALDH* expression in both the roots and leaves. The expression of *ZmALDH* only responded to Al toxicity but not to other stresses including low pH and other metals. The heterologous overexpression of *ZmALDH* in *Arabidopsis* increased Al tolerance by promoting the ascorbate-glutathione cycle, increasing the transcript levels of antioxidant enzyme genes as well as the activities of their products, reducing MDA, and increasing free proline synthesis. The overexpression of *ZmALDH* also reduced Al accumulation in roots. Taken together, these findings suggest that *ZmALDH* participates in Al-induced oxidative stress and Al accumulation in roots, conferring Al tolerance in transgenic *Arabidopsis*.

## 1. Introduction

Aluminum (Al) is one of the most abundant metal elements in the Earth’s crust and can be solubilized to the toxic species Al^3+^ in acidic soils (pH < 5.0) [1]. Al stress, which can rapidly inhibit root elongation even at a micromolar concentration, is one of themajor factors limiting crop yield in acidic soils [2]. Improving the Al tolerance of crops in acidic soils has been a longstanding aim for solving food shortages and improving biofuel production.

In the past decades, two kinds of aluminum tolerance mechanisms, organic acid extradition for external detoxification and Al sequestration into vacuoles for internal tolerance, have been adopted by plants [3,4,5]. In particular, the secretion of organic acids from the root apices, especially malate, citrate, and oxalate, is a widely used strategy in plants [1,6,7].

Al stress induces oxidative stress, which causes the overproduction of reactive oxygen species (ROS) and aldehydes in plants [8,9,10]. Aldehydes are common intermediates in most cellular metabolic pathways in plants and can be produced under various environmental stresses, such as cold, heat shock, salinity, and desiccation [11,12]. The excessive accumulation of ROS and aldehyde substances could seriously destroy the cell membrane, change the fluidity and ion transport characteristics of the biomembrane, hinder protein synthesis, and eventually result in cell death [13,14,15]. Plants could resist the Al-induced ROS by enhancing antioxidant systems [16,17]. The ascorbate-glutathione (AsA-GSH) cycle can directly scavenge H_2_O_2_ produced in plants, and antioxidants produced in the AsA-GSH cycle (including AsA and GSH) can remove ROS [18,19]. The AsA-GSH cycle is also involved in the redox balance by the regulation of four key enzymes, including APX (ascorbate peroxidase), MDAR (monodehydroascorbate), DHAR (dehydroascorbate reductase), and GR (glutathione reductase) [20]. The detoxification of aldehydes depends on the activity of aldehyde dehydrogenases (ALDH, EC 1.2.1.3) [21,22]. ALDHs are ubiquitous in both prokaryotes and eukaryotes. They belong to the NAD(P)^+^-dependent protein superfamily, which catalyzes the oxidation of aldehydes to the corresponding non-toxic carboxylic acids and balances the content [23]. Based on sequence identity and substrate specificity, ALDHs could be organized into two main groups and at least 12 subfamilies in plants [11,24]. Previous studies have revealed that ALDH could function as a primary aldehyde dehydrogenase to cope with various environmental stresses by preventing and protecting plants against oxidative stress [15,25,26]. For example, OsALDH2B1, which functions primarily as an aldehyde dehydrogenase, not only protects against various stresses, but regulates the growth and reproduction in rice [26]. *Ath-ALDH3* could increase the tolerance to drought, salt, and heavy metal (Cu^2+^ and Cd^2+^) stress in transgenic *Arabidopsis* [14]. *Ath-ALDH3I1* and *Ath-ALDH7B4* confer tolerance to osmotic and oxidative stress and protect plants against lipid peroxidation [15]. *ZmALDH7B6* was induced by multiple environmental stresses and might play a role in aldehyde detoxification [25]. In addition, *ALDH* genes might be also involved in the response to Al toxicity, since the expression of an *ALDH2* homolog was up-regulated in a resistant genotype of highbush blueberry (*Vaccinium corymbosum*) under Al stress [27].

In our previous transcriptomic analysis of maize under Al stress, *ZmALDH* (*Zm00001d017418*), an *ALDH* gene family member, was up-regulated by Al [28]. However, the physiological mechanism by which this gene contributes to the response to Al stress is unclear. Here, the role of the *ZmALDH* gene in resistance to Al stress was investigated by analyzing Al-induced oxidative stress, including the roles of the AsA-GSH cycle and antioxidant enzyme system as well as overexpression assays.

## 2. Results

### 2.1. Cloning and Sequence Analysis of ZmALDH

We obtained the complete *ZmALDH* CDS via PCR amplification from the Al-resistance maize inbred line 178 (Appendix A) [29]. The *ZmALDH* gene is 1458 bp long with nine exons and eight introns (Appendix A). It encodes a 446 aminoacid transmembrane protein with a predicted molecular mass of 49.81 kDa and isoelectric point of 6.40 (Appendix A). The deduced protein sequence showed sequences identified at 70.16% and 72.28% with rice aldehyde dehydrogenases OsALDH3-1 and OsALDH3-2, respectively (Figure 1A). A 98 amino acid sequence between D71 to D168 and a 236 amino acid sequence between W167 to V402 formed two Aldedh domains, which are involved in oxidoreductase activity.

In a phylogenetic analysis, ZmALDH clustered with several members of the Class 3 family of ALDHs, including OsALDH3-1, OsALDH3-2, AtALDH3F1, AtALDH3, AtALDH3H1, and GmALDH3H2 (Figure 1B) [14].

### 2.2. Expression Analysis of ZmALDH

To better understand the physiological function of *ZmALDH*, its expression patterns in different organs were examined by semi-quantitative RT-PCR. As shown in Figure 2A, *ZmALDH* was expressed in all tested organs, including the root, stem, leaf, tassel, ear, and seed, indicating that *ZmALDH* influences the growth and development of maize. To identify whether the transcription of *ZmALDH* was induced by Al stress at the seedling stage, its expression was measured by quantitative RT-PCR. The results revealed that *ZmALDH* was expressed in both leaves (L) and roots, and the expression level at the root tips (RT, apical 0–1 cm) was consistently higher than that in the rest of root (ROR) under Al stress (Figure 2B). The expression of *ZmALDH* increased initially until reaching a peak at 12 h and then decreased thereafter in the roots and leaves. A dose–response curve analysis showed that the expression of *ZmALDH* was induced by low concentrations of Al (10 μM) and increased until 60 μM AlCl_3_, after which expression levels were maintained at a steady level (Figure 2C).

To determine the Al-specificity of *ZmALDH* upregulation, the effects of various divalent and trivalent cations were also evaluated in the roots. As shown in Figure 2D, the expression of *ZmALDH* in the roots did not respond to low pH, cadmium (Cd), lanthanum (La), zinc (Zn), copper (Cu), manganese (Mn), or iron (Fe), except for Al.

### 2.3. Subcellular Localization

Prediction algorithms indicated that ZmALDH targets the cytoplasm or the chloroplast (Appendix A). To determine its subcellular localization, we performed a transient assay with the *ZmALDH-GFP* fusion gene under the control of the CaMV*35S* promoter (35S:*ZmALDH::GFP*) in tobacco leaves and maize protoplasts. The control vector consisted of the CaMV*35S* promoter driving the expression of GFP alone (pCAMBIA2300-*eGFP*). As shown in Figure 3 and Appendix A, for the control vector, soluble GFP was localized to the cytoplasm and the nucleus in both cell types, as expected. However, the fluorescence of the 35S:*ZmALDH::GFP* chimera was associated with the cell and the nuclear periphery of the maize protoplasts and tobacco leaf cells, suggesting a plasma and nuclear membrane localization.

### 2.4. Root Growth Analysis of ZmALDH Transgenic Arabidopsis

To investigate the function of *ZmALDH*, the coding region of *ZmALDH*, driven by the CaMV *35S* promoter, was stably introduced into wild-type (WT) *Arabidopsis*. Two highly expressed homozygous T_3_ lines (L4 and L6) were identified by PCR and RT-PCR (Appendix A), and used in subsequent analyses.

To determine whether *ZmALDH* overexpression could enhance Al tolerance in transgenic *Arabidopsis*, we examined the growth of the two transgenic lines (L4 and L6) and WT plants treated with 0, 50, and 100 μM AlCl_3_. In the absence of Al, the root growth of all plants was similar (Figure 4A,B). After treatment with 50 μM AlCl_3_, the relative root elongation (RRE) values for L4 and L6 were 78.96% and 73.92%, respectively, compared with 44.23% in WT (Figure 4C). At a high concentration of Al (100 μM), the RRE of all tested plants further decreased further (Figure 4C), but was still higher in the transgenic lines than in WT plants, implying that the root growth was more resistant to Al toxicity in transgenic plants than in WT (Figure 4).

### 2.5. ZmALDH Overexpression Reduces Al Accumulation

We further evaluated whether the increased resistance to Al toxicity observed in the transgenic lines (L4 and L6) was related to Al accumulation. We first visualized the Al uptake in roots by staining with hematoxylin, where darker blue-purple staining indicates greater Al absorption. In control conditions, staining in the root, including the root tip and basic root, was very weak. However, staining became darker after Al treatment (Figure 5A). Compared to the intensity in WT plants, L4 and L6 had weaker staining intensities, especially at the root tips (Figure 5A). Further, we quantified these differences by measuring the Al content by ICP-MS. In the absence of Al, WT, L4, and L6 plants did not exhibit any discernible differences and contained very low concentrations (~2.5 mg/g DW) of Al (Figure 5B). The Al contents of these three plants increased after exposure to Al, however, Al accumulation in WT plants was more than 1.8-fold higher than that in L4 and L6 plants. Al contents in leaves were similar in all materials and were not affected by Al treatment (Appendix A).

### 2.6. Comparative Analysis of the Oxidative Stress Response under Al Treatment

To determine oxidative damage induced by Al toxicity, we measured the production of H_2_O_2_ and the superoxide anion radical (O_2_^−^). DAB staining was used to visualize the production of H_2_O_2_, where darker staining indicates more H_2_O_2_ production in the tissue. As shown in Figure 6A,B, WT, L4, and L6 plants contained a similarly low concentration of H_2_O_2_, approximately 1.13 μmol g^−1^ FW in the absence of Al. The H_2_O_2_ concentration increased in all plants after exposure to Al, whereas H_2_O_2_ in WT plants was 1.7-fold higher than that of L4 and L6 (Figure 6B). Similar to H_2_O_2_, the concentrations of O_2_^−^ increased in all plants after Al treatment, whereas L4 and L6 accumulated only 60% of the O_2_^−^ levels in the WT (Figure 6C). These results confirmed that the accumulation of ROS induced by Al stress in transgenic lines was less than that in WT plants.

Excessive production and accumulation of ROS in plants would directly trigger membrane lipid peroxidation, resulting in cell damage or death [10,30]. Therefore, we further measured the absorption of Evans blue and the accumulation of malondialdehyde (MDA). Levels of Evans blue uptake after 12 h of Al treatment in the root tips of WT plants were approximately 3-fold higher than those in L4 and L6 (Figure 7A,B). Meanwhile, the MDA contents in WT, L4, and L6 were similar to those in the control condition. After 12 h of Al stress, the accumulation of MDA in WT plants was 1.8-fold higher than that in L4 and L6 (Figure 7C). These results suggested that the overexpression of *ZmALDH* in *Arabidopsis* plants could reduce lipid peroxidation and cell damage caused by Al stress.

### 2.7. ZmALDH Enhanced the Tolerance to Oxidative Stress Induced by Al Toxicity

NAD(P)H, producedby ALDH catalytic oxidative, is an important substance for the maintenance of the balance of glutathione (GSH) in plants. The latter was indispensable for the AsA-GSH cycle, which is necessary to remove ROS effectively [19]. To investigate whether the increased Al tolerance of transgenic lines was related to the AsA-GSH cycle, we detected the contents of two antioxidants, including GSH and AsA. As shown in Figure 8, the GSH and AsA contents were maintained at a low level in the absence of Al. However, after exposure to 60 μM AlCl_3_ for 12 h, the GSH accumulation in WT plants and two transgenic lines increased 16.2%, 73.5%, and 66.5%, respectively, and AsA increased to 537 μg/g FW, 918 μg/g FW, and 922 μg/g FW, respectively. These results implied that the overexpression of *ZmALDH* could improve the contents of antioxidants associated with the AsA-GSH cycle.

Subsequently, GR and APX activity levels were measured. The activities of GR and APX increased upon Al exposure, and the levels in transgenic L4 and L6 plants were approximately 70% and 200%, respectively, greater than in the WT plants (Figure 9A,B). Consistent with these findings, a real-time PCR analysis confirmed that the transcript levels of *AtGR1* and *AtAPX1* were about 1.29- and 2.5-fold higher, respectively, in L4 and L6 transgenic lines than in WT plants after exposure to Al (Appendix A). These results indicated that the Al tolerance of *ZmALDH* transgenic lines was related to the increased activity and expression levels of GR and APX in the AsA-GSH cycle.

The ROS scavenging ability is also linked to another enzymatic system. Hence, we further determined the transcript levels of the ROS scavengers as well as the activities of their products. The activities of SOD, POD, and CAT in WT, L4, and L6 plants were similar in the absence of Al. Under Al stress, activity levels of these ROS-scavenging enzymes were higher in L4 and L6 (by 50%, 100%, and 130%, respectively) than in the WT (Figure 10A–C). In addition, the transcript levels of *AtSOD1* and *AtPOD1* showed similar patterns in all lines in control conditions and were up-regulated in transgenic lines after exposure to Al (Figure 10D,E). However, the *AtCAT1* expression trend was similar to that of *AtAPX1* in WT and transgenic lines, regardless of Al treatment (Figure 10F). These results demonstrated that *ZmALDH* could reduce the oxidative damage induced by Al stress by promoting ROS scavenging.

### 2.8. ZmALDH Promoted the Accumulation of Cellular Osmolytes

Plant ALDHs contribute to the synthesis of osmolytes, including sorbitol, myo-inositol, and proline, which have hydroxyl radical scavenging activity [31,32]. Therefore, we detected the proline content in WT and transgenic plants under control and Al stress conditions. As shown in Figure 11, the proline contents in all plants were similar under control conditions. After Al treatment, the levels of proline accumulation in L4 and L6 were 1.4-fold higher than those in WT plants, indicating that the overexpression of *ZmALDH* increased the accumulation of proline, potentially increasing Al tolerance in transgenic *Arabidopsis*.

## 3. Discussion

The ALDH superfamily is a group of enzymes able to catalyze the conversion of aldehydes to the corresponding acids [12,21]. *ALDH* genes are activated by environmental stresses [14,15,25] and have a certain influence on the stress response. *VCAL68*, a homolog of *ALDH*, is the only reported gene involved in the response to Al stress in plants [27]. However, the functions of *ALDH* genes in the response to Al stress are not well-characterized. In the present study, we cloned and characterized an aldehyde dehydrogenase gene from maize, *ZmALDH*, and investigated the role of *ZmALDH* under Al stress by its overexpression in *Arabidopsis*.

*ZmALDH* belongs to class 3 ALDHs, which are involved in the detoxification of aldehydes [14]. It had two Aldedh domains and was highly homologous to *OsALDH3-1* and *OsALDH3-2* (Figure 1), suggesting that *ZmALDH* exerts a similar function to that of this gene subfamily. In normal conditions, *OsALDH3-1* and *OsALDH3-2* are highly expressed in the young leaf and stem, with little expression in the young root [33]. However, *ZmALDH* was expressed in almost all tissues, with high expression in the root, stem, leaf, and ear (Figure 2A). This difference may be attributed to interspecific difference, or gene function evolution. Previous studies have shown that *OsALDH3-1* and *OsALDH3-2* are down-regulated by both drought and high salinity stresses [33]. Moreover, the maize genome contains ~28 *ALDH* genes, and the transcript level of *ZmALDH* is repressed by drought stress but strongly induced by salt stress [24,34]. In our study, *ZmALDH* was specifically up-regulated by Al stress and peaked at 12 h after Al treatment (Figure 2). This differed from the expression pattern of *VCAL68*, which exhibited expression peaks at 2 h and 24 h under Al stress [27]. These findings suggested that the response of *ALDH* genes to different stresses differs among plant species.

The subcellular localization indicates functional specialization. Plant ALDHs show wide variation in localization; for example, AtALDH3I1, AtALDH7B4, and OsALDH2B1 are located in chloroplasts, cytoplasmic [15], and nuclear compartments [26], respectively. However, ZmALDH in this study was localized in both the plasma membrane and nuclear membrane (Figure 3), different from any previous ALDHs. We speculated that Al-induced oxidative stress on the one hand leads to membrane lipid peroxidation and the production of toxic substances (such as ROS and aldehydes) [35,36], thereby activating the expression of *ALDH* genes and stimulating the synthesis of antioxidant enzymes and antioxidants, which were transported to the damaged plasma membrane to alleviate the oxidative stress [15,33]. On the other hand, an increase in the Al tolerance of transgenic lines was also related to a decrease in the Al content in roots, which is affected by the expression of *ZmALDH*. Thus, *ZmALDH* might have an influence on the crosstalk between the Al accumulation and oxidative stress response mechanisms.

Several studies have shown that Al-induced oxidative stress is an early process in toxicity responses in plants [9,37,38]. In this study, Evans blue and DAB staining revealed that Al stress caused plasma membrane damage and H_2_O_2_ accumulation (Figure 6 and Figure 7). ALDHs could decrease oxidative stress by scavenging hydroxyl radicals and cytotoxic aldehydes [39,40]. Similar to the *Ath-ALDH3* response to cadmium stress [14], the contents of ROS (H_2_O_2_ and O_2_^−^) and MDA in *ZmALDH* transgenic *Arabidopsis* decreased (Figure 6 and Figure 7), weakening of the oxidative stress induced by Al toxicity. Furthermore, under Al stress, the transgenic *Arabidopsis* with *ZmALDH* overexpression exhibited higher Al tolerance, including longer roots and less Al accumulation (Figure 5). This is perhaps due to the efficient conjugation of GSH with Al and/or ROS via overexpression of *ZmALDH*. Consequentially, the overexpressed transgenic plants showed lower Al content, lower ROS content, and less oxidative damage under Al stress. This is quite similar with the case of *OsGSTU6* which improved the Cd tolerance of transgenic rice accompanied with low Cd content via ensuring the efficient conjugation of GSH with Cd and/or ROS [41]. These results suggested that *ZmALDH* contributed to Al tolerance and had a potential role in the detoxification of reactive aldehydes derived from lipid peroxidation.

Coincidentally, overexpression of some Al-inducible ROS-mediated and/or -related genes could promote Al tolerance [42]. For example, the overexpression of alternative oxidase gene (*AOX*) and *MnSOD1* enhanced Al resistance in tobacco cells [43] and transgenic *Brassica napus* [44], respectively. Transgenic *Arabidopsis* plants overexpressing *AtPrx64* exhibited tolerance to Al stress [45]. Overexpression of *ZmAT6* in both maize and *Arabidopsis* could partially enhance Al tolerance via scavenging ROS [46]. Additionally, enhancing the activity of antioxidases in antioxidant pathways also contributes to resistance to Al toxicity in plants [47]. In our study, *ZmALDH* enhanced the Al tolerance of *Arabidopsis* by promoting the synthesis of GSH and AsA, enhancing the enzymatic activity of GR and APX, and increasing the transcript level of *AtGR1* (Figure 8, Figure 9 and Appendix A). Furthermore, overexpression of *ZmALDH* could also increase the expression levels of *AtSOD1* and *AtPOD1*, and improved the activity of these two antioxidant enzymes (Figure 10).

Under abiotic stress conditions, such as salt, drought [48], and Al stress [49], higher proline accumulation protects cells via osmotic adjustment and toxic free radical scavenging [50]. We obtained similar results, in which proline accumulation was slightly higher in transgenic lines than in WT plants (Figure 11). Together, these findings demonstrated that the overexpression of *ZmALDH* in *Arabidopsis* conferred tolerance to Al stress by enhancing its oxidative stress tolerance.

## 4. Materials and Methods

### 4.1. Plant Materials, Growth Conditions, and Treatment

The Al-tolerant maize inbred line 178 [29] and *Arabidopsis thaliana* (Columbia ecotype, Col-0) were used in this study. Seed treatment, germination, and culture conditions were performed as described in a previous study [46].

The metal treatments, time-course analysis, and manipulation of Al concentrations were performed as described previously [51]. The roots and/or the shoots from all the treated seedlings were immediately frozen in liquid nitrogen, and stored at −80 °C for RNA extraction.

For Al tolerance analysis of transgenic *Arabidopsis*, two-week-old WT plants and transgenic lines grown in 2% MGRL solution (pH 4.5) [50] were pre-treated with 0.5 mM CaCl_2_ for 30 min and then treated with or without 60 μM AlCl_3_ for 12 h. Root and shoot samples were collected, separately.

### 4.2. Cloning and Sequence Analysis of ZmALDH

Based on the reference sequence of B73 cDNA, the *ZmALDH* gene was amplified from 178 by specific primers (Appendix A). A multiple sequence alignment of amino acid sequences (Appendix A) was generated using DNAMAN (Lynnon). A phylogenetic tree was inferred using the neighbor-joining method by MEGA 6.0 [52].

### 4.3. RNA Isolation and Gene Expression Analysis

Total RNA extraction, semiquantitative RT-PCR and real-time PCR were performed as described previously [53]. *ZmGAPDH* and *AtACT2* were used as the internal reference genes. Three biological replicates were performed for each experiment. The primers are listed in Appendix A.

### 4.4. Subcellular Localization

The subcellular localization of the ZmALDH protein was predicted using several online websites (Appendix A) and determined by transient expression in tobacco leaves and maize leaf protoplasts with a 35S:*ZmALDH::GFP* translational fusion cloned in the pCAMBIA2300 vector [46,54].

### 4.5. Generation of Transgenic Arabidopsis Plants

The p35S:*ZmALDH* vector was transformed into *Arabidopsis* ecotype Columbia-0 (Col-0) by the flowering dip method [55]. The transgenic plants were selected as described previously [56]. Finally, the positive homozygous seeds of the T_3_ generation were used for further experiments.

### 4.6. Measurement of Oxidative Damage

Oxidative stress induced by Al stress was evaluated by cell death and membrane lipid peroxidation. The former was estimated by determining Evans blue uptake [57] and the latter was detected by the concentration of malondialdehyde via the thiobarbituric acid method [8].

### 4.7. Determination of Reactive Oxygen Species (ROS)

The production of H_2_O_2_ was visualized by 3,3′-diaminobenzidine (DAB) staining according to previously described methods [17] and measured as described before [46]. The super oxygen free radical (O_2_^−^) was detected by the hydroxylamine method [58].

### 4.8. Determination of Glutathione and Ascorbic Acid

The total GSH and AsA contents were determined according to methods of Hodges et al. [59] and Noctor and Foyer [60].

### 4.9. Enzymatic Activity and Gene Expression Analyses

Antioxidant enzyme extraction and the measurement of SOD, POD, CAT, and APX activity levels were carried out as previously described [10,46]. GR activity was determined by the decrease in absorbance at 340 nm due to NADPH oxidation [61].

The relative expression levels of enzyme genes described above were measured by RT-PCR, and the primers are listed in Appendix A.

### 4.10. Measurement of Relative Root Growth

The sterilized seeds were first grown in 1/2 MS medium (pH 4.5) for 3 days. Then the seedlings were transplanted into 1/8 MS medium (pH 4.5) with or without 50 μM and 100 μM AlCl_3_ for 7 days, and the initial and final root lengths were recorded. The RRE was defined as the ratio of the net root growth length after treatment with Al to that of the control.

### 4.11. Al Content Assay

The accumulation of Al was visualized by hematoxylin staining [62]. The Al contents in WT and transgenic plants with or without Al treatment were measured by ICP-MS (PerkinElmer, NexlON 2000, Seer Green, UK).

### 4.12. Statistical Analysis and Reproducibility

All treatments were repeated at least three times. Statistical analyses, including Tukey’s test, were performed using SPSS 21.0 (IBM, Chicago, IL, USA, 2012).

## 5. Conclusions

*ZmALDH* could promote the Al tolerance of plants by enhancing the AsA-GSH cycle, increasing the transcript levels of antioxidant genes as well as activating their products, and mediating the accumulation of Al in roots.

## Figures and Tables

**Figure 1 ijms-23-00477-f001:**
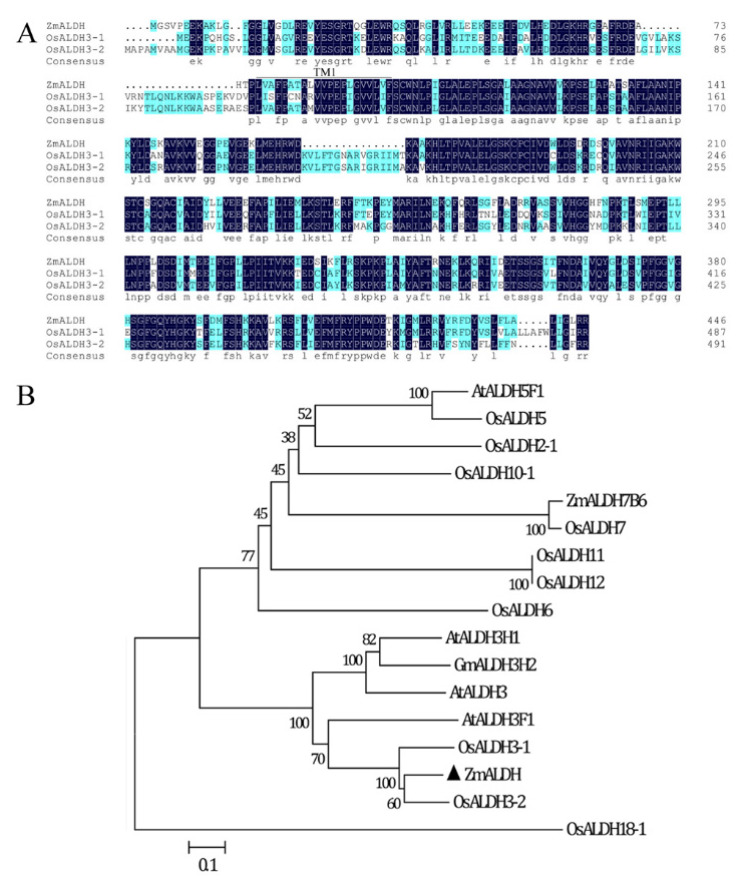
Sequence and phylogenetic analysis of ZmALDH and other known ALDH proteins: (**A**) Multiple sequence amino acid alignment (DNAMAN) of ZmALDH (AQK73130), OsALDH3-1 (XP_015623875), and OsALDH3-2 (XP_015623591); (**B**) Phylogenetic relationships of ZmALDH and previously published ALDH proteins. Unrooted NJ trees were generated using MEGA 6.0. Bootstrap values from 1000 replicates are indicated at each branch.

**Figure 2 ijms-23-00477-f002:**
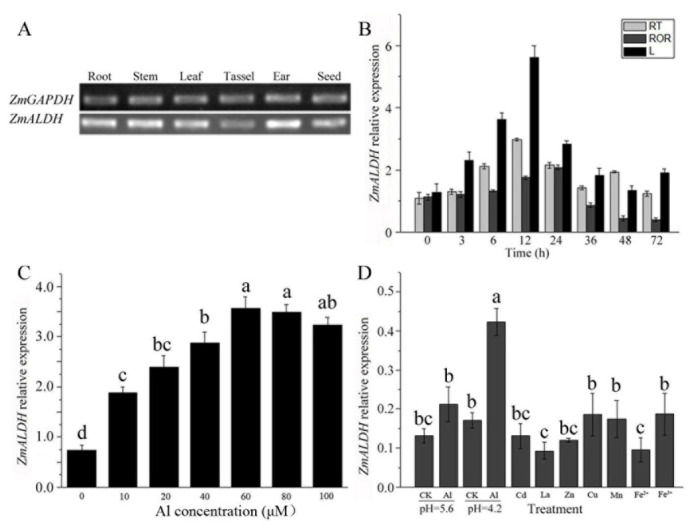
Expression analysis of the *ZmALDH* gene: (**A**) Spatial expression patterns of *ZmALDH* in different tissues; (**B**) Transcript levels of *ZmALDH* in the maize seedlings of root tips (RT), the rest of the root (ROR), and the leaves (L) were quantified at 0, 3 h, 6 h, 12 h, 24 h, 36 h, 48 h, and 72 h after Al treatment (pH 4.2); (**C**) Expression analysis of *ZmALDH* under different concentrations of Al. Seedlings were treated for 6 h with 0 μM, 10 μM, 20 μM, 40 μM, 60 μM, 80 μM, and 100 μM Al, and then the roots were collected for RT-PCR; (**D**) Effect of other cations on ZmALDH expression. Seedlings were exposed for 6 h to a 0.5 mM CaCl_2_ solution (pH 4.2) containing 30 μM CdCl_2_, 2.0 μM CuCl_2_, 10 μM LaCl_3_, 100 μM ZnCl_2_, 200 μM MnSO_4_, 20 μM FeSO_4_·7H_2_O, 20 μM FeCl_3_, or 60 μM AlCl_3_, respectively. The roots were then collected for subsequent RT-PCR. Relative gene expression normalized to the *ZmGAPDH*. The values are presented as mean ± SD (*n* = 5) and different letters indicate significant difference at *p* < 0.05 (Tukey’s test).

**Figure 3 ijms-23-00477-f003:**
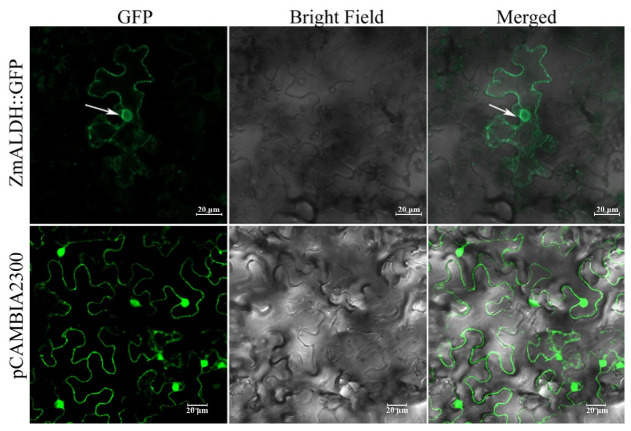
Subcellular localization of ZmALDH in tobacco leaves.

**Figure 4 ijms-23-00477-f004:**
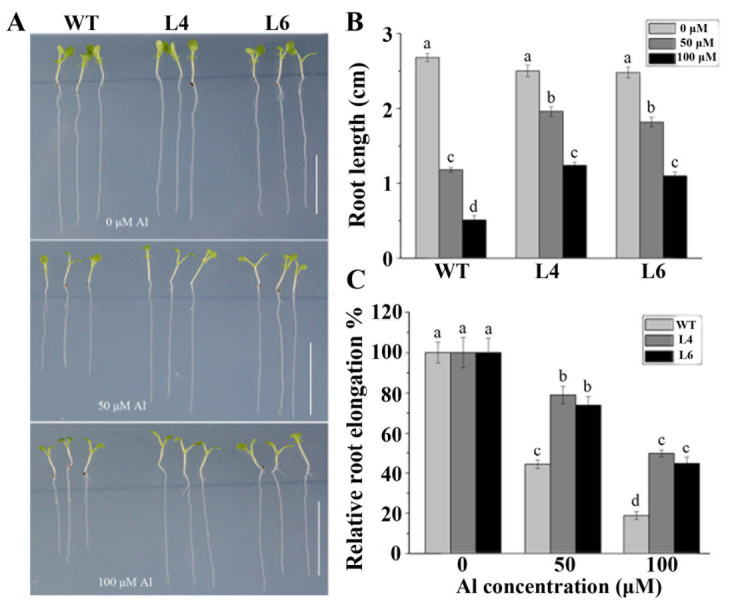
Transgenic plants expressing *ZmALDH* are more resistant to Al: (**A**) Phenotype; (**B**) Root length, and (**C**) Relative root elongation of WT and transgenic lines under normal and Al stress conditions. Seedlings of WT and two *ZmALDH* transgenic lines (L4 and L6) were grown on a plate containing 0 μM, 50 μM, and 100 μM AlCl_3_ for 5 d. Scale bar: 1.0 cm. Values represent mean ± SD (*n* ≥ 15). Different letters indicate significant differences (*p* < 0.01) (Tukey’s test).

**Figure 5 ijms-23-00477-f005:**
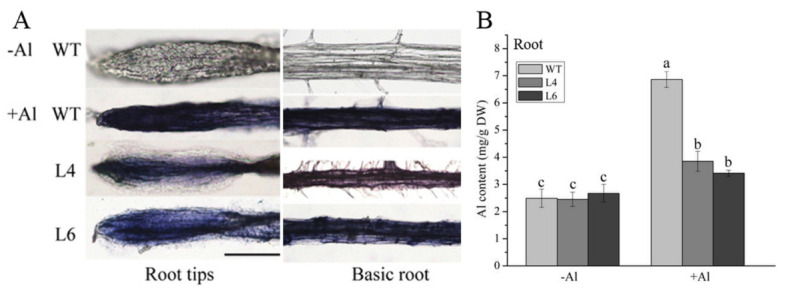
Al accumulation in roots is reduced in transgenic *Arabidopsis*. (**A**) Hematoxylin staining after 12 h exposure to 0 and 60 μM AlCl_3_ in roots. Scale bar: 50 μm; (**B**) Al contents in lines (as in (**A**)) under Al stress for 12 h. Values represent mean ± SD (*n* ≥ 15). Different letters indicate significant differences (*p* < 0.01) (Tukey’s test).

**Figure 6 ijms-23-00477-f006:**
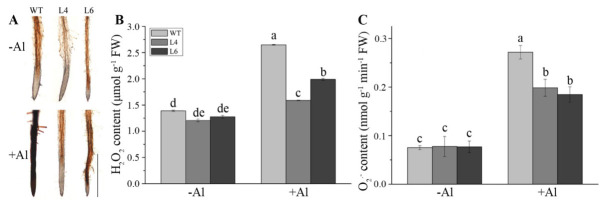
Overexpression of *ZmALDH* reduces the accumulation of superoxide (O_2_^−^) and hydrogen peroxide (H_2_O_2_). Seedlings of WT, L4, and L6 were grown in the 2% MGRL solution for 14 d, and then plants were treated with 60 μM AlCl_3_ for 12 h, before roots were collected for subsequent experiments: (**A**) Histochemical visualization of H_2_O_2_ was performed with DAB staining. Scale bar: 250 μm. Contents of (**B**) H_2_O_2_ and (**C**) O_2_^−^ in WT and transgenic lines after exposure to Al stress for 12 h. Values represent mean ± SD (*n* = 20). Different letters indicate significant differences (*p* < 0.01) (Tukey’s test).

**Figure 7 ijms-23-00477-f007:**
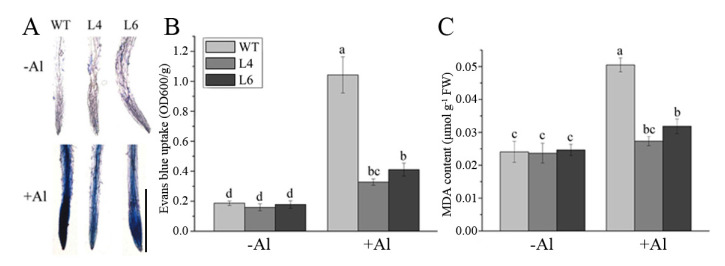
Effect of Al treatment on the lipid peroxidation and loss of plasma membrane integrity in roots. Seedling roots of wild–type plants and transgenic lines were collected after exposure to 60 μM AlCl_3_ for 12 h and used to evaluate (**A**) Evans blue staining, scale bar: 200 μm; (**B**) Evans blue uptake, and (**C**) Malondialdehyde (MDA) contents. Values represent mean ± SD (*n* = 20). Different letters indicate significant differences (*p* < 0.01) (Tukey’s test).

**Figure 8 ijms-23-00477-f008:**
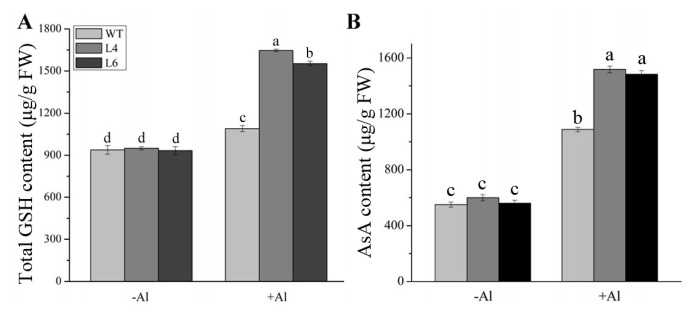
Overexpression of *ZmALDH* increases the accumulation of total GSH and AsA in roots. The total contents of (**A**) GSH and (**B**) AsA in WT and transgenic *Arabidopsis* (L4 and L6) were detected after exposure to 60 μM AlCl_3_ for 12 h. Values represent mean ± SD (*n* = 20). Different letters indicate significant differences (*p* < 0.01) (Tukey’s test).

**Figure 9 ijms-23-00477-f009:**
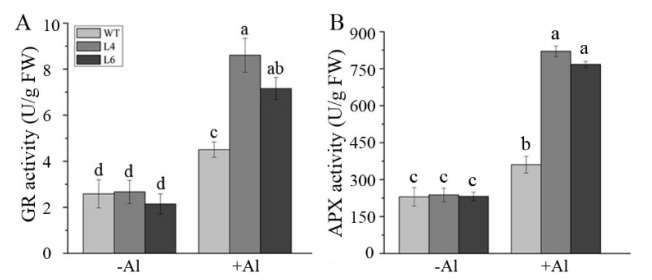
Activity of GR and APX in wild–type and transgenic plants. The activity of (**A**) GR and (**B**) APXin wild–type plants and transgenic lines after exposure to Al for 12 h. Values are means ± SD (*n* = 20). Different letters indicate significant differences (*p* < 0.01) (Tukey’s test).

**Figure 10 ijms-23-00477-f010:**
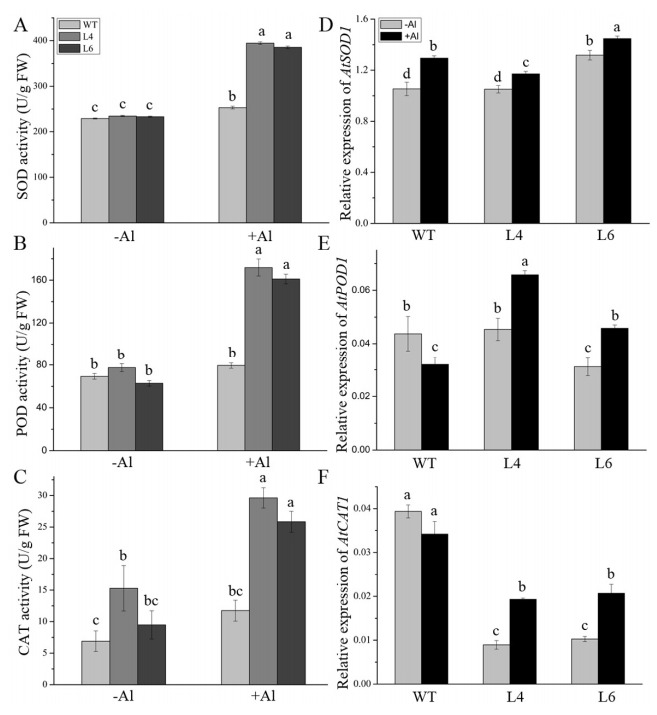
Effect of Al on the activities and transcripts’ level of antioxidant enzymes. Two–week–old seedlings were treated with 60 μM AlCl_3_ for 12 h and the total activity of (**A**) SOD; (**B**) POD, and (**C**) CAT and transcripts’ level of (**D**) *AtSOD1*; (**E**) *AtPOD1*, and (**F**) *AtCAT1* were analyzed. Relative gene expression was normalized to the *AtACT2*. Values are means ± SD (*n* = 20). Different letters indicate significant differences (*p* < 0.05) (Tukey’s test).

**Figure 11 ijms-23-00477-f011:**
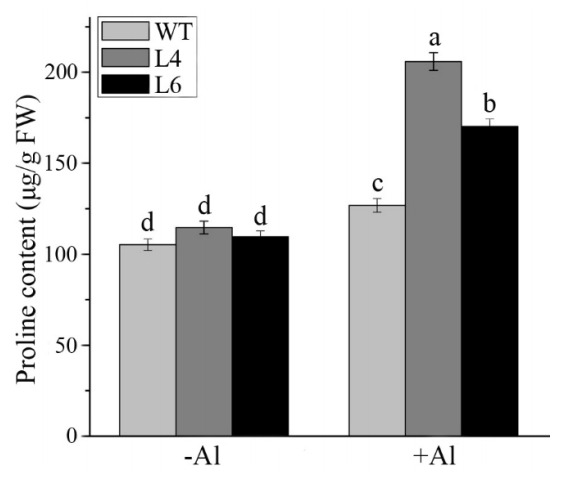
Determination of free proline in wild–type plants and transgenic lines. Two-week-old seedlings were treated with or without 60 μM AlCl_3_ for 12 h, and the proline contents of roots were determined after that. Values are means ± SD (*n* = 20). Different letters indicate significant differences (*p* < 0.05) (Tukey’s test).

## Data Availability

All data generated or analyzed during this study are available within the article or upon request from the corresponding author.

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
