# Peer review of "Overexpression of the Aldehyde Dehydrogenase Gene ZmALDH Confers Aluminum Tolerance in Arabidopsis thaliana"

_ijms, 2022, doi:10.3390/ijms23010477_

Round 1

Reviewer 1 Report

Dear Authors,

Congratulations on your work on the aldehyde dehydrogenase gene ZmALDH expression in Arabidopsis plants. The research plan is complete and very well described. I have made a few comments that I think could help improve furthermore your work. 

  1. Line 114 : (RT)
  2. Line119: 200 μM MnSO4, 20 μM FeSO4·7H2O,
  3. Line 121: Turkey’s test
  4. To determine the Al-specificity of ZmALDH upregulation, the effects of various divalent and trivalent cations were also evaluated in the roots. Question: was this analyzed with the same pH value?
  5. 2. Expression analysis of ZmALDH : Specify better in text and picture subtitle in which organs the gene was analyzed.
  6. Line 135: maize protoplasts
  7. Figure 3: make the scale bars bigger. You can’t see them!!
  8. Figure S3b Y axis : the relative expression of ZmALDH
  9. 4. Root growth analysis of ZmALDH transgenic Arabidopsis. Question: Is the line expressing or overexpressing??? how can you state that the plants are overexpressing the gene rather than just expressing it?
  10. 5. ZmALDH overexpression reduces Al accumulation. Question: why was this analysis done at 60 micromolar concentration and not at 50 or 100 micromolar Al? please add an explanation also in the text.
  11. Figure 5: what does the scale bar correspond to? Is it the same for all the images? Please state it, if so.
  12. Figure 6C Y axis: nmol/g/min FW? Is min correct?
  13. 6. Comparative analysis of the oxidative stress response under Al treatment. Question: at what concentration was this analysis done?
  14. Line 282, please correct: It had two Aldedh domains
  15. ZmGAPDH and AtACT2: Why were the reference genes used separately and for two different gene analysis?
  16. Line 376: change the font size
  17. 1. Plant materials, growth conditions, and treatment: Please write down the conditions in which you have grown Arabidopsis plants in this section. In the papers, you have mentioned as a reference, the growing conditions are for maize plants. You did not perform only root measurements on Arabidopsis, but all the other analysis too.
  18. Please be more precise on the seedling age (two weeks = 15 days) and day or hours of treatment per type of analysis.

Author Response

Dear Editor and reviewers:

Thank you so much for your helpful comments on our manuscript. Meanwhile, we would like to express our sincere appreciation to the anonymous reviewers for their carefully reading and invaluable comments. We have revised the manuscript in accordance with the reviewers’ comments, and carefully proof-read the manuscript to minimize typographical, grammatical, and bibliographical errors. Here below is our description on revision according to the reviewers’ comments.

Responses to the comments of reviewer 1

  • Line 114 : (RT)

Line 119: 200 μM MnSO4, 20 μM FeSO4·7H2O,

Line 121: Turkey’s test

Reply: Thank you for your comments. We have corrected these minor errors. And the correction could be seen in the revised manuscript (Line 113-122).

  1. To determine the Al-specificity of ZmALDH upregulation, the effects of various divalent and trivalent cations were also evaluated in the roots. Question: was this analyzed with the same pH value?

Reply: Thank you for your professional comments. Both the treatment and the analysis were carried out based on pH 4.2.

  1. Expression analysis of ZmALDH: Specify better in text and picture subtitle in which organs the gene was analyzed.

Reply: Thanks a lot for your comments. We have added detailed information to the picture subtitle in the revised manuscript (Line 116 and 120).

  1. Line 135: maize protoplasts

Reply: We have corrected the description as you suggested.

  1. Figure 3: make the scale bars bigger. You can’t see them!!

Reply: Thank you for your suggestion. We modified the scale bar that we can see it clearly now.

  1. Figure S3b Y axis : the relative expression of ZmALDH

Reply: Thank you. We have modified it according to your suggestion.

  1. Root growth analysis of ZmALDH transgenic Arabidopsis. Question: Is the line expressing or overexpressing??? how can you state that the plants are overexpressing the gene rather than just expressing it?

Reply: Thank you for your professional comments. These are overexpression lines (L4 and L6). We had constructed a plant expression vector containing the full length of ZmALDH gene, and transformed it into wild-type Arabidopsis by Agrobacterium infection. The transgenic plants were detected by PCR and the positive plants were obtained. Then we detected the expression of ZmALDH in the positive plants by real-time fluorescence quantitative RT-PCR. The expression of ZmALDH was very high in both L4 and L6 (Fig.S3). Therefore, we believed that these two lines which we used in this study are overexpressed lines.

  1. ZmALDH overexpression reduces Al accumulation. Question: why was this analysis done at 60 micromolar concentration and not at 50 or 100 micromolar Al? please add an explanation also in the text.

Reply: Thank you for your comments. Firstly, when we analyzed the expression of ZmALDH under different concentration of Al in maize, we found that its expression was the highest under 60 μM AlCl3. So, this concentration was also used in the following study of transgenic Arabidopsis. Secondly, the medium and nutrient solution are different. 50 or 100 micromolar Al were the concentration used in 1/8 MS medium which was used for root growth analysis. Therefore, we didn’t use these two concentrations in these studies.

  1. Figure 5: what does the scale bar correspond to? Is it the same for all the images? Please state it, if so.

Reply: The scale bar here corresponds to the ratio of root stained with hematoxylin in Fig 5A. We have added the information in the revised manuscript (L176).

  1. Figure 6C Y axis: nmol/g/min FW? Is min correct?

Reply: In fact, it is correct. The “min” here represents the amount of superoxide (O2·-) produced per min.

  1. Comparative analysis of the oxidative stress response under Al treatment. Question: at what concentration was this analysis done?

Reply: This analysis was evaluated under 60 μM AlCl3 treatment for 12 h. we have added this information in the revised manuscript (L192-194 and L209-212).

  1. Line 282, please correct: It had two Aldedh domains.

Reply: The writing of Aldedh domain is correct. We refer to this paper (Zhao et al., 2015) here.

Zhao, C.H.; Wang, D.; Feng, B.; Gou, M.; Liu, X.; Li, Q.W. Identification and characterization of aldehyde dehydrogenase 9 from Lampetra japonica and its protective role against cytotoxicity. Comp Biochem Phys B 2015, 187, 102-109.

  1. ZmGAPDH and AtACT2: Why were the reference genes used separately and for two different gene analysis?

Reply: Thanks to your professional comments. Because the species used in the real-time fluorescence quantitative RT-PCR are different (maize and Arabidopsis), two reference genes are used. ZmGAPDH is commonly used in maize and AtACT2 is usually used in Arabidopsis.

  1. Line 376: change the font size

Reply: We have changed the font size in the revised manuscript (L384-386).

  1. Plant materials, growth conditions, and treatment: Please write down the conditions in which you have grown Arabidopsis plants in this section. In the papers, you have mentioned as a reference, the growing conditions are for maize plants. You did not perform only root measurements on Arabidopsis, but all the other analysis too.

Reply: Thank you for your comments. We added the detail information in the revised manuscript (L354-357).

  1. Please be more precise on the seedling age (two weeks = 15 days) and day or hours of treatment per type of analysis.

Reply: We have added more information in the revised manuscript to better understand the culture and treatment time of seedlings as you suggested.

Reviewer 2 Report

Dear Colleagues

I read the received manuscript with an interest. The topic of this research is relevant. The manuscript contains new data indicating that overexpression of the ZmALDH gene of aldehyde dehydrogenase in maize is accompanied by an increase in the tolerance of Arabidopsis plants to the toxic effect of aluminum. The authors cloned and sequenced the gene and performed phylogenetic analysis of the protein encoded by this gene with other known ALDHs. The organ, tissue and subcellular localization of ALDH was investigated. The main attention in the work is paid to the proof that overexpression of the ZmALDH gene in Arabidopsis plants leads to an increase in aluminum tolerance, primarily due to a decrease in the level of oxidative stress. This is achieved through the activation of antioxidant enzymes and the accumulation of a number of organic antioxidants. Data were also obtained indicating that overexpression of the ZmALDH gene is accompanied by a decrease in the supply of aluminum ions to the root cells, which also leads to a decrease in the generation of reactive oxygen species.
The data obtained by the authors are quite convincing, although I will express some considerations below.
1. A little confused by the fact that the changes in all measured indicators fully corresponded to the idea of ​​the authors of this article. This happens rarely. We will assume that in this case the model turned out to be adequate for solving the problem posed. At the same time, it is not often possible to meet a situation when all investigated cellular antioxidant systems are simultaneously activated. Most often, the opposite happens, part of the defense systems under stress is activated, while the other part does not change or even inhibited.
2. Finally, the authors write “We obtained similar results, in which proline accumulation was much greater in transgenic lines than in WT plants (Fig. 11)”. From my point of view, the observed increase in proline level under Al stress conditions is very small. Its intracellular content has practically no osmoregulatory effect due to its low concentration in the cell; the antioxidant effect of proline in this case is also highly questionable.
3. I believe that the authors should try to explain how the overexpression of the ZmALDH gene leads to a decrease in the intake of aluminum into the root cells. This effect requires comment.
In general, I believe that after minor revisions, the manuscript can be recommended for publication in IJMS. This research has been carried out quite thoroughly.

Author Response

Dear Editor and reviewers:

Thank you so much for your helpful comments on our manuscript. Meanwhile, we would like to express our sincere appreciation to the anonymous reviewers for their carefully reading and invaluable comments. We have revised the manuscript in accordance with the reviewers’ comments, and carefully proof-read the manuscript to minimize typographical, grammatical, and bibliographical errors. Here below is our description on revision according to the reviewers’ comments.

Responses to the comments of reviewer 2

  1. A little confused by the fact that the changes in all measured indicators fully corresponded to the idea of the authors of this article. This happens rarely. We will assume that in this case the model turned out to be adequate for solving the problem posed. At the same time, it is not often possible to meet a situation when all investigated cellular antioxidant systems are simultaneously activated. Most often, the opposite happens, part of the defense systems under stress is activated, while the other part does not change or even inhibited.

Reply: Thank you for your professional comments. We have also considered these results. Compared with the previous studies, both antioxidant enzymes and antioxidants play a certain role in our study, which may be related to the fact that the gene encodes aldehyde dehydrogenase that plays an important role in AsA-GSH cycle. Additionally, although the action trend of several enzymes is similar, the activity variation of the enzymes is different. For example, the activity of SOD is more than 10 times that of CAT. Therefore, we believe that the functions of each enzyme are still different.

  1. Finally, the authors write “We obtained similar results, in which proline accumulation was much greater in transgenic lines than in WT plants (Fig. 11)”. From my point of view, the observed increase in proline level under Al stress conditions is very small. Its intracellular content has practically no osmoregulatory effect due to its low concentration in the cell; the antioxidant effect of proline in this case is also highly questionable.

Reply: Thank you for your comments. The levels of proline accumulation in L4 and L6 were 1.4-fold higher than those in WT plants after exposure to Al. Therefore, we think it plays a role in osmotic regulation. We have modified the description “much greater”of this sentence into “slight higher” in the revised manuscript (L342).

  1. I believe that the authors should try to explain how the overexpression of the ZmALDH gene leads to a decrease in the intake of aluminum into the root cells. This effect requires comment.
    In general, I believe that after minor revisions, the manuscript can be recommended for publication in IJMS. This research has been carried out quite thoroughly.

Reply: Thank you for your professional suggestion. We agree with you at this point since we had not provided experimental data to explain the decrease content of Al in the root of OE plants. This is perhaps due to the efficient conjugation of GSH with Al and/or ROS via overexpression of ZmALDH. Consequentially, the overexpressed transgenic plants showed lower Al content, lower ROS content and less oxidative damage under Al stress. There are some articles that have described the similar phenomenon. For example, the higher expression level of OsGSTU6 had improved the Cd tolerance of transgenic rice companied with low Cd content via ensuring the efficient conjugation of GSH with Cd and/or ROS (Jing et al., 2020). In addition, overexpression of a peroxidase gene AtPrx64 in tobacco alleviated the oxidative stress induced by Al stress and also reduced the Al accumulation in roots (Wu et al., 2017). We have added relevant information in the “Discussion” of the revised manuscript (L319-325).

Jing, X.Q.; Zhou, M.R.; Nie, X.M.; Zhang, L.; Shi, P.T.; Shalmani, A.; Miao, H.; Li, W.Q.; Liu, W.T.; Chen, K.M. OsGSTU6 contributes to cadmium stress tolerance in rice by involving in intracellular ROS homeostasis. J Plant Growth Regul 2021, 40, 945-961.

Wu, Y.S.; Yang, Z.L.; How, J.Y.; Xu, H.N.; Chen, L.M.; Li, K.Z. Overexpression of a peroxidase gene (AtPrx64) of Arabidopsis thaliana in tobacco improves plant’s tolerance to aluminum stress. Plant Mol Biol 2017, 95, 157-168.